# An Exploratory Study among Intellectual Disability Physicians on the Care and Coercion Act and the Use of Psychotropic Drugs for Challenging Behaviour

**DOI:** 10.3390/ijerph181910240

**Published:** 2021-09-29

**Authors:** Janouk C. bij de Weg, Aline K. Honingh, Marieke Teeuw, Paula S. Sterkenburg

**Affiliations:** 1Mungra Medical Centre, Annastraat 35, Nieuw-Nickerie, Suriname; janoukbijdeweg@gmail.com; 2Faculty of Behavioural and Movement Science, Vrije Universiteit Amsterdam, 1081 BT Amsterdam, The Netherlands; a.k.honingh@vu.nl; 3Ben Sajet Centrum, 1011 JH Amsterdam, The Netherlands; marieketeeuw@gmail.com; 4Bartiméus, Assessment and Treatment, 3941 XM Doorn, The Netherlands

**Keywords:** psychotropic drugs, challenging behaviour, intellectual disability, medical legislation, physicians

## Abstract

The new Dutch Care and Coercion Act aims to better regulate the use of psychotropic drugs for challenging behaviour in people with an intellectual disability. This study explores experiences of intellectual disability physicians (IDPs) in prescribing psychotropic drugs and investigates how the Act and the new multidisciplinary guideline on challenging behaviour affects their practice. A qualitative study was conducted, consisting of nine semi-structured in-depth interviews with IDPs, followed by a thematic analysis. It was found that IDPs experienced the new Act and guideline as supportive of their work as guardians of the appropriate use of psychotropic drugs. The multidisciplinary character of the guideline was experienced positively. However, IDPs are faced with organisational barriers and time constraints, as such, they question the feasibility of implementing the Act. Based on these findings, it can be concluded that the Care and Coercion Act may support the existing shift towards the appropriate use of psychotropic drugs if required conditions can be met.

## 1. Introduction

Intellectual disability physicians (IDPs) are often faced with requests to prescribe psychotropic drugs to treat challenging behaviour of people with an intellectual disability. It appears that 30% to 50% of children and adolescents with an intellectual disability also have mental disorders [1]. Furthermore, there is a greater risk for children and adolescents with an intellectual disability of developing behavioural problems [2], such as antisocial and self-absorbed behaviour [3]. The prevalence of challenging behaviour, such as disruptive or destructive behaviour, aggression and self-injury, is as high as 10% to 15% among people with an intellectual disability [4,5]. Challenging behaviour is often the primary reason for the prescription of psychotropic drugs for people with an intellectual disability [6], even though little evidence exists that these medications are effective in these situations [7,8]. Furthermore, they are often used off-label, which means that they are not prescribed in accordance with the registered indication [6,9,10]. The prevalence of antipsychotic drug use in a Dutch population living in sheltered care facilities was 32.2%; furthermore, in 58% of these cases, challenging behaviour was the primary reason for the prescription of antipsychotic drugs [6]. The prevalence of psychotropic drugs prescribed to people with intellectual disabilities in the United Kingdom was found to be respectively 37.7% [11], 49.1% [12] and 49% [13]. Minimising the use of restrictive practices in the care of people with an intellectual disability has been on the international political agenda for many years [14,15]. Policy changes and legislative reforms in various countries aim to reduce both physical and chemical (the use of psychotropic drug) restraints. Nevertheless, the actual practice is not always in line with these ambitions [16,17]. Moreover, side effects from the use of psychotropic drugs, such as concentration problems, emotional indifference, fatigue and adverse metabolic effects, are common in populations of people with intellectual disabilities [18]. It is therefore important to reduce the unnecessary prescription of psychotropic drugs.

In January 2020, new Dutch legislation regarding involuntary care and coercive measures, the Care and Coercion Act (in Dutch: Wet zorg en dwang), entered into force. The new legislation, which replaces the Psychiatric Hospitals (Compulsory Admissions) Act (in Dutch: Wet Bijzondere opnemingen psychiatrische ziekenhuizen, Bopz), aims to regulate the rights of people with an intellectual disability, as well as people with a psychogeriatric disorder, when receiving involuntary care. ‘No compulsion, unless…’, is the core of the new Act. Involuntary care can only be applied if there is no other option [19]. This Act also applies to the administration of psychotropic drugs if these are not prescribed in accordance with applicable guidelines (i.e. ‘off-label’). Even if a client shows no resistance and/or the representative gives consent, a procedure must be followed to minimise the use of these medications. The new legislation will therefore have consequences for the daily practice of IDPs, who have expressed concerns about not being sufficiently prepared for the implementation of the new legislation because of the anticipated high workload, shortages of staff and administrative burdens [20].

Causes of challenging behaviour are complex and sometimes multifactorial [21]. Challenging behaviour can be related to physical complaints, environmental, psychological or social factors, psychiatric disorders and other behavioural factors [6,21,22]. Although it is recommended to only prescribe off-label psychotropic drugs for a short period of time [23], it is often continued for more than ten years [24]. Reasons for IDPs to prescribe or continue off-label antipsychotic drugs include aggression, restlessness, autism spectrum disorders, pressure from family, support workers or managers [9], previously unsuccessful discontinuation trajectories [24] and ‘other reasons’ (among these, lack of appropriate environmental circumstances).

In October 2019, just before the enforcement of the new legislation, a new ‘multidisciplinary guideline challenging behaviour in adults with an intellectual disability’ [25] was published. It provides guidance concerning the multidisciplinary approach of challenging behaviour by strengthening non-pharmacotherapeutic interventions and the vigilant and effective use of psychotropic drugs. In the guideline, alternative ways of treatment are suggested, e.g., the Triple-C intervention [26] and Integrative Therapy for Attachment and Behaviour [27]. The guideline refers to the stepped care plan of the new Act as a legally required tool to ensure that all options for alternative care are considered when creating a multidisciplinary treatment plan.

The motivation of the current study is threefold: 1. psychotropic drugs are prescribed for a longer period than is recommended [24]; 2. the new Dutch Care and Coercion Act applies to the administration of psychotropic drugs and will have an impact on the daily practice of IDPs; and 3. IDPs expect a higher workload and extra administrative actions and have, therefore, expressed their concerns about the implementation of the legislation [20]. This study, therefore, aims to explore IDPs’ expectations and experiences concerning the new Act and the new multidisciplinary guideline with respect to the prescription of psychotropic drugs. The research questions are: 1. how do IDPs experience or expect the new legislation to influence their practice; and 2. which facilitators or barriers do they anticipate or experience with the arrival of the new Act and multidisciplinary guideline. Findings will mark the IDPs’ perspectives on the new legislation and the use of psychotropic drugs for challenging behaviour and can shed light on the concerns that IDPs have and problems that they face with respect to the implementation of the new Act.

## 2. Materials and Methods

### 2.1. Design

This study fits into descriptive phenomenology, where a qualitative exploratory research design was conducted, using a thematic analysis approach. The thematic analysis was chosen for this study because this method provides the possibility of gaining knowledge about the experiences and key issues by describing various recurring or common themes [28,29].

### 2.2. Participants

Dutch IDPs who provide care for people with an intellectual disability in sheltered care facilities and/or outpatient clinics were approached. Interviews were conducted with nine IDPs who were all female and worked in five provinces of the Netherlands. Their average age was 46 years, ranging between 31–63 years. The participants had an average work experience as an IDP of 9 years, ranging between 2 months and 19 years. IDP was recognised as an independent specialisation in 2000. The IDPs worked in different settings: outpatient clinics, group home care and/or specialised day care centres, covering different severities of intellectual disability. 

### 2.3. Procedure and Data Collection

Due to the COVID-19 outbreak, there were two recruitment phases. In the first phase, two IDPs in training made a selection of IDPs working in sheltered care facilities, divided over the twelve provinces of the Netherlands. A list of twenty-seven IDPs was collected, aimed at reaching IDPs who had a variation in work experience and work environment (with respect to size, region and training of IDPs). We strived for a broad range with respect to age and years of work experience and aimed for variety with respect to work environment, in order to ensure a sample that was representative of the population of IDPs in the Netherlands. The diversity of the sample contributes to the transferability of the study. The first author sent an e-mail to twenty-seven IDPs with an invitation in which the purpose of the study was mentioned and the informed consent form was included. Eight IDPs responded but two participants dropped out. In the second recruitment phase, a psychiatrist and a developmental psychologist provided information on another fourteen IDPs for recruitment. Four IDPs responded but one participant dropped out. See Figure 1 for an overview of the recruitment process.

The Scientific and Ethical Review Board (VCWE 2020-007) of the Faculty of Behaviour and Movement Sciences, Vrije Universiteit Amsterdam, approved the research proposal. Before the start of the interviews, all participants gave their written informed consent. Before the COVID-19 lockdown (from March until May 2020) four interviews were conducted in the office of the IDPs and, after the lockdown, five interviews were conducted by telephone. Eight interviews were conducted by J.W. and one by M.T. and lasted for about 45 minutes each. Data saturation was reached after eight interviews; thus, in the eight interviews no new information was reported. To be sure about the data saturation, a ninth interview was conducted, but no additional information was gained. The fact that data were collected until data saturation was reached contributes to the credibility of the study. The interviews were recorded with a voice-recorder and were then transcribed.

The topic guide contained specific questions about age and work experience, general questions about work as an IDP and open-ended questions about: 1. experiences with prescribing psychotropic drugs; 2. the influence of the new legislation on prescribing psychotropic drugs; and 3. the value of the new guideline in their daily practice when prescribing psychotropic drugs. By using open-ended questions, the IDPs could elaborate on points that were important to them, so that all relevant issues were included.

### 2.4. Data Analysis

A thematic analysis was used to analyse the data. The thematic analysis was data driven and explorative, rather than theoretically driven (deductive analysis), and themes were identified inductively, so as to work towards broader generalisations and theories [29]. The thematic analysis was based on a step-by-step guide containing six phases outlined by Braun and Clarke [30]. In the first step of the analysis, two researchers (M.T. and J.W.) familiarised themselves with the data by reading and re-reading all interviews. In this phase, initial notes were made about ideas on interesting aspects of the data. In the second step M.T. and J.W. independently coded the interviews. M.T. used post-it notes and J.W. used the software program ATLAS.ti 8.4 for coding. As many potential themes as possible were coded. As it was an inductive approach, the emerging codes were discussed until agreement was reached on the interpretation of the data [29]. In the third step, a set of candidate themes was devised, and a mind map was used as a visual representation. In the fourth step, the themes were reviewed and refined, and the thematic map was adjusted. In the fifth step, a third researcher (P.S.) became part of the analysing process and the (sub)themes were discussed. In the final step, the report was produced, including a description of each theme and the subthemes, accompanied by quotes from the interviews. A member check was performed by asking all participants for feedback on the results [31]. Their comments were discussed by two researchers (M.T. and J.W.) and minor adjustments were made.

## 3. Results

Three major themes with corresponding subthemes were identified, as displayed in Figure 2. The three major themes were: 1. the multidisciplinary approach; 2. optimising the use of psychotropic drugs; and 3. transitional phase.

The quotes accompanying the theme descriptions were translated from Dutch. The translations were kept as close as possible to the original words that were used, without deviating from the meaning of a sentence.

### 3.1. The Multidisciplinary Approach

#### 3.1.1. The Multidisciplinary Team

IDPs agreed that a multidisciplinary team should treat challenging behaviour. Before the new Act came into force, IDPs already worked closely together with colleagues from various disciplines. This view is in agreement with the multidisciplinary guideline. With regard to the treatment of a client, IDPs keep an overview and know which specialisms need to be engaged. The following people are mentioned as having a role in these multidisciplinary collaborations: client, family or legal representative, IDP, developmental psychologist, support worker, (paediatric-)psychiatrist, nurse, general practitioner, psychomotor therapist, physiotherapist, occupational therapist, speech therapist, movement counsellor and pastoral caretaker. Depending on the situation, IDPs rely on information from the client, team members and family. The roles of the developmental psychologist and support worker have been most frequently mentioned as important team members. At outpatient locations, however, the involvement of a developmental psychologist is not standard and contact with the care team is more difficult since team members are spread over various locations. In the Care and Coercion Act, it is mandatory that several designated parties from the multidisciplinary team are involved when involuntary care is applied following the stepped care plan. Depending on the step in the stepped care plan, there will be new multidisciplinary collaborations, for instance with an external expert. 

The involvement of a psychiatrist on a consultation basis differs between organisations, mostly depending on the setting—sheltered care facility or outpatient—and the severity of the intellectual disability. Two IDPs reported permanent involvement of a psychiatrist. Five IDPs can consult a psychiatrist when needed, some of whom do so regularly. Most consultations involve clients with a mild intellectual disability, since psychiatrists have less experience with severe intellectual disabilities than IDPs. The role of the psychiatrist in the multidisciplinary team was said to be mostly for diagnosing a psychiatric disorder or to give advice on psychotropic drugs. However, the availability of psychiatrists is often a problem. Moreover, organisations face shortages of professional staff, which seems to be an even bigger problem in the outpatient setting.
‘Challenging behaviour should never be treated by a single person, you should always do it together’ (12 years’ work experience as an IDP).
‘I do think that the psychiatrist generally knows more about the effect of psychotropic drugs in the brain. And whether or not it is safe to increase the dose’ (2 months’ work experience as an IDP).
‘Well, I would really like the cooperation with the psychiatrist to improve, the consultations being easily accessible’ (9 years’ work experience as an IDP).

#### 3.1.2. The New Guideline as Facilitator

The new multidisciplinary guideline contains recommendations for professionals involved in identifying and treating challenging behaviour. The guideline advises on support and treatment of challenging behaviour, on the prescription of psychotropic drugs and on the composition of the team depending on the part of the treatment. The IDP is not necessarily involved from the start of the treatment. Knowledge of the content of the new guideline varies among IDPs, from ‘having a general idea of the content’ (six IDPs), to ‘being familiar with the content’ (two IDPs), to ‘having cooperated in the development and revision’ (one IDP).

IDPs in general feel that their current and past practice of working in a multidisciplinary team is already in line with the recommendations of the new guideline. The multidisciplinary aspect of the new guideline is perceived as positive by five IDPs (the others did not comment explicitly on the multidisciplinary character). They stressed the importance of multidisciplinary collaboration when it comes to analysing and treating challenging behaviour.
‘[I] thought of multidisciplinary consultation sooner and questioned the developmental psychologist more: “Do you have measurement lists or diagnostic lists for this? Could it be this and that too?” [I] questioned a little more (…) also think somewhat broader and also questioned the staff involved more’ (5 years’ work experience as an IDP).
‘[It] Really helps to professionalise the organisation, (…) the multidisciplinary approach and who does what at what time and who is responsible for something’ (7 years’ work experience as an IDP).

### 3.2. Optimising the Use of Psychotropic Drugs

#### 3.2.1. Role Perception of the IDP

IDPs have a clear perception of their role when it comes to diagnostics and treatment of challenging behaviour. They perceive the appropriate treatment of challenging behaviour as one of their core tasks and strengths. In order to make a (pharmacological) treatment plan, they perceive a thorough case investigation as essential: medical files are inspected to learn about past assessments and past psychotropic drug prescriptions and their effects. In case of unclear diagnosis, extra assessments can be performed, or a psychiatrist can be consulted.

Challenging behaviour without a psychiatric diagnosis is usually not an indication for prescribing psychotropic drugs and, therefore, can only be prescribed off-label. The general attitude of IDPs is to be conservative in prescribing off-label psychotropic drugs, but they report this is not always the general attitude in the multidisciplinary team. Lack of knowledge, previous experiences and the severity of challenging behaviour can lead to the desire to prescribe or not phase out psychotropic drugs amongst family, support workers and developmental psychologists. IDPs reported seeing an important role for themselves for transferring their knowledge about psychotropic drugs and challenging behaviour. IDPs felt that there are sometimes high expectations of both family and staff when it comes to the effect of psychotropic drugs, especially in an ambulatory setting. There are certain situations in which IDPs prescribe psychotropic drugs more frequently, for instance, in emergency situations, where this medication is meant to be administered for a short period only. Furthermore, if psychotropic drugs are indicated and considered effective for a specific diagnosis, IDPs have no problem with prescribing psychotropic drugs, such as for clients diagnosed with autism and challenging behaviour.
‘What I always do is dive into a case completely and see what has been done and what has been prescribed’ (12 years’ work experience as an IDP).
‘We as IDPs are being trained in integral thinking, not only psychotropic drugs, but also about physical complaints, the interaction between physical and mental health. And [IDPs] also better know what developmental psychologists and support workers can and should be able to do’ (7 years’ work experience as an IDP).
‘Well, I think, the thoughts about psychotropic drugs within sheltered care facilities and the idea that it can help a lot has been a continuous battle. I have been on a mission for years to explain to everybody that psychotropic drugs are for psychiatric disorders’ (8 years’ work experience as an IDP).

#### 3.2.2. The New Act as Facilitator

According to the Care and Coercion Act, the stepped care plan is always mandatory for medication that influences a client’s behaviour or freedom of movement and is not administered in accordance with applicable guidelines. This even applies if the client is incapacitated, shows no resistance and the representative gives consent.

IDPs reported the new legislation as being supportive and helpful in the optimisation of the use of psychotropic drugs. Some clients are using psychotropic drugs because of an automatic continuation of a prescription by another doctor in the past. Because of the new legislation, several IDPs have now reassessed the use of psychotropic drugs and when not prescribed in accordance with the registered indication, started the stepped care plan. One IDP reported that she had already phased out psychotropic drugs for a number of clients as a result of the evaluation.
‘[I] actually even reduced and stopped [the prescription], because when I looked, I Thought “what are we doing here, it may be that side effects are now the cause of the challenging behaviour, for which a measure is now being applied”’ (19 years’ work experience as an IDP).
‘Now you are being pushed into that stepped care plan and you just have to look at it more often and you may have to use a little more creative thinking and even get someone from outside to “just think along”. Maybe we are looking, are we already in such a tunnel that we overlook certain things’ (16 years’ work experience as an IDP).

#### 3.2.3. Guidelines as Facilitators

Multiple guidelines exist to advise IDPs. It is not mandatory to use a guideline for the treatment of clients, but if psychotropic drugs are not prescribed in accordance with a guideline, the stepped care plan, defined in the Care and Coercion Act, needs to be followed.

The new ‘multidisciplinary guideline challenging behaviour in adults with an intellectual disability’ contains a section on indications for psychotropic drugs. IDPs said that they use the multidisciplinary guideline mostly for this section, since the new legislation has influenced the prescription of psychotropic drugs. The full version of the new guideline is reported to be unfeasible for use in daily practice because of the length. IDPs suggested the guideline could be more compact and concrete.

Besides the new multidisciplinary guideline, other guidelines to support IDPs exist as well. The guideline Prescribing Psychotropic Drugs from the Netherlands’ Society of Intellectual Disability Physicians (in Dutch: NVAVG) and the psychiatry guidelines are used most often by IDPs. Other guidelines that are mentioned are the psychiatry guidelines for autism, ADHD, anxiety and depression, as well as the child-and-youth psychiatry guidelines. IDPs claimed to choose a guideline based on the diagnosis, if there was one. If there was no diagnosis (yet), psychotropic drugs were usually not indicated and could therefore only be prescribed off-label. The NVAVG guideline Prescribing Psychotropic Drugs and the new multidisciplinary guideline would be used in these cases. These guidelines make suggestions for the treatment of challenging behaviour but emphasise the off-label character (and therefore the necessity of following the stepped care plan) of prescribing psychotropic drugs.
‘I think if we want to use it, maybe we should get some sort of a maximum of three, four pages, otherwise the implementation will also be difficult’ (16 years’ work experience as an IDP).
‘Well, often I know that there is a certain diagnosis, either autism or an anxiety disorder, I take the psychiatry guidelines, because they really focus on that. And if I think of someone with addiction with no clear diagnosis, or auto mutilation for example, not much is known about it. Then I actually take the other guideline [NVAVG guideline]’ (3 years’ work experience as an IDP).

### 3.3. Transitional Phase

#### 3.3.1. Current Concerns

As a consequence of the Care and Coercion Act, extra workload for IDPs seems inevitable. IDPs expressed their concerns about extra tasks that they have to take up according to the Act. For example, the Care and Coercion Act mentions the role of an external expert, who should be asked for advice if off-label psychotropic drugs could not be phased out after a designated period. IDPs are expected to take up this role and to serve as external experts for other organisations. Furthermore, IDPs also expressed their concerns related to the administrative burden that comes with the Care and Coercion Act. Moreover, two IDPs expressed concerns about how the exchange of external experts between institutions will be organised.

There is great variation between organisations in the extent that the Care and Coercion Act has been implemented. In one organisation, all off-label psychotropic drugs were reassessed, while some other organisations had not taken any steps to implement the Act yet. Other differences between organisations that were noted included the way that external experts were exchanged and the method for organising training and information sessions about the new legislation.

Six IDPs mentioned that how to implement the Act was not entirely clear to them; for instance, with respect to the adjustments needed to register involuntary care in the electronic patient file and the training the team would receive. Furthermore, in some organisations there is still uncertainty about the deployment of external experts, both in group-home care settings and outpatient clinics. One IDP mentioned that the organisational aspects of the Care and Coercion Act in home settings is still completely unclear. Additionally, the new multidisciplinary guideline has not been fully implemented yet. Although the guideline is known among IDPs, it is still unknown to most support workers.
‘There is simply a shortage of IDPs, no one has time and it [external expert role] is not something you can easily do extra, in a right way’ (…) ´Because what I now understood is mainly a paper idea of those external experts. I am afraid it will become a “paper tiger”’ (12 years’ work experience as an IDP).
‘Unfortunately, we do not yet have any agreements with other organisations [for the external expert], which should be there by now’ (2 months’ work experience as an IDP).
‘No, unfortunately not [not much known by support workers]. We agreed on an implementation process when it was published, but the person who has to do that remains anxiously silent. (…), so I am afraid that we [the organisation] will be delayed’ (2 months’ work experience as an IDP).

#### 3.3.2. Future Perspectives

Due to the new legislation, IDPs expect to increasingly receive questions about off-label psychotropic drugs from developmental psychologists and support workers in the team. Answering these questions and educating the team is, together with the extra task of acting as an external expert, adding up to the workload of the IDP. It is questioned by IDPs if the new legislation is feasible for them because of the extra workload.

IDPs acknowledge that, in the past, when less information on psychotropic drugs was available, psychotropic drugs were prescribed too often and sometimes for too long a period. In recent years, however, a shift is being seen that is in line with the Care and Coercion Act. Doctors prescribe less psychotropic drugs than they used to and also (relatives of) clients show a growing awareness of the side effects of psychotropic drugs.

If organisations manage to implement the Care and Coercion Act properly, IDPs expect improvement in the quality of care. One IDP mentioned that she hopes the new legislation will serve as support for her choice to be conservative in prescribing certain medication. The clear rules of the Act can possibly convince other team members and relatives more easily.
‘I do expect that we will get many more questions about psychotropic drugs. Especially when one finds out that there is actually quite a lot still off-label. (…) I do expect that there will be a huge increase in questions for our profession. (…) I don’t think so [being ready for this as a profession]’ (2 months’ work experience as an IDP). 
‘And I have noticed a very big shift in recent years towards less prescription of psychotropic drugs in general. So that’s a very positive trend. (…) I really think professionalisation of care. (…) where you see that where historically psychiatrists reached for psychotropic drugs faster, IDPs do so less quickly’ (7 years’ work experience as an IDP).
‘So you are going to be even more careful about that [steps] and you will be extra tested by that external expert, first internally and then externally. So I think you can get even more careful in the end. (…) I really think that can improve the quality’ (9 years’ work experience as an IDP).

## 4. Discussion

This study aimed to investigate the practice of IDPs’ prescription of psychotropic drugs and explored IDPs’ expectations and experiences concerning the new Act and multidisciplinary guideline. It was found that both the new Act and guideline facilitate the treatment of challenging behaviour in a multidisciplinary setting with IDPs in their aspired role as guardians of the appropriate use of psychotropic drugs. The multidisciplinary character of the guideline was experienced as positive. However, IDPs are faced with organisational barriers, time constraints and question the feasibility of the implementation of the Act.

Challenging behaviour was said to be ideally treated in a multidisciplinary team. This multidisciplinary collaboration is experienced to be more difficult in outpatient clinic settings, which is a problem since the government aims to promote extramural care in the home environment [32]. The IDPs reported that there is less knowledge and higher expectations of the effects of psychotropic drugs in the outpatient setting. The government’s choice to promote extramural care in the home environment is therefore at odds with the care that can be provided by the IDPs for people with intellectual disabilities and challenging behaviour. In outpatient care, multidisciplinary treatment of challenging behaviour by IDPs, psychiatrists, professional support workers and developmental psychologists should be the norm, according to the IDPs.

The general attitude of the IDPs towards prescribing psychotropic drugs is conservative. This can be at odds with the fear of phasing out psychotropic drugs amongst family, support workers and developmental psychologists caused by, for instance, lack of knowledge. Nevertheless, a finding of our study is a shift towards less prescription of psychotropic drugs by IDPs. By working in a multidisciplinary way, there are more alternatives for treatment such as providing better matched caregiving [26] or psychotherapy [27].

The findings of shortages in staff and the administrative burden of the Care and Coercion Act are in line with the reported concerns by the Dutch Association for the care of people with disabilities [20]. It is questioned by IDPs whether the implementation of the Care and Coercion Act is feasible due to these two barriers. Despite these barriers, with respect to the implementation, IDPs are positive about the content of the Care and Coercion Act and already experienced or expect the new legislation to support them in the appropriate use of psychotropic drugs.

With respect to the experienced barriers, certain recommendations can be formulated. The implementation of the Care and Coercion Act is found to be handled individually by each organisation. An example is the deployment of external experts. It is worth discussing whether a regional or national approach from the beginning of the implementation would have been supportive for the organisations in order to anticipate future regulations on a national level.

Furthermore, although IDPs were positive about the multidisciplinary aspect of the new guideline, the guideline was experienced as being too lengthy and therefore difficult to apply in daily practice. IDPs suggested that a compact version would be more practical. In the most recent issue of the Dutch journal for intellectual disability physicians (*Tijdschrift voor Artsen voor Verstandelijk Gehandicapten*), a summary of the new guideline was published [33]. This summary can increase knowledge about the content of the guideline, although the target audience of this journal (IDPs) already has some knowledge of this. In contrast, support workers were found to be less familiar with the guideline and may therefore benefit from a summary of the guideline in a journal or magazine targeted at them.

Three final recommendations, stemming directly from the results, are as follows: first, with respect to the shortages of staff, both the availability of psychiatrists and IDPs should be improved. Second, the availability of developmental psychologists and support workers for a multidisciplinary collaboration in the outpatient setting should be improved. Third, the treatment team should have a better knowledge of the effect of psychotropic drugs and the new multidisciplinary guideline. 

Limitations of our study are as follows: first, because of the lockdown period related to COVID-19, the present study had two interview periods instead of one. It is possible that participants who were interviewed after the lockdown period had a different perspective towards the Act and the guideline than participants who were interviewed before. We have looked into this and found no difference between the interviews before and after the lockdown. Second, the small sample size of nine all-female participants, partly due to the difficulties in recruitment with the COVID-19 outbreak, should be acknowledged. In the Netherlands, the majority of IDPs are female: 80.6% [34]. Although data collection was continued until data saturation was achieved, this means in this case that the results may be generalisable to the population of female IDPs only. It is important that, in future research, male IDPs should be heard on these topics as well. Third, the interviews were conducted and analysed in Dutch, but the report was written in English. The quotes used for the report were translated to English and although we saw no evidence for lacking nuances or possibly contradictory findings, language issues still may have influenced the quotes. However, participants were sent the results in order to be able to check them and they did not consider the results, and thereby the translated quotes, to be misinterpreted.

## 5. Conclusions

This is the first qualitative study that explored both the experiences of IDPs with the new legislation and guideline and the subsequent influence on their daily practice. From this study, it can be inferred that the Care and Coercion Act supports the already existing shift towards appropriate use of psychotropic drugs, if required conditions can be met. The results give insight into the implementation of the Care and Coercion Act and, subsequently, the barriers faced by IDPs. This study should be considered as a first exploration on these topics. More research is needed to see whether the opinions of IDPs on the Care and Coercion Act and the multidisciplinary guideline are shared by psychiatrists, developmental psychologists, support workers, clients and their family.

## Figures and Tables

**Figure 1 ijerph-18-10240-f001:**
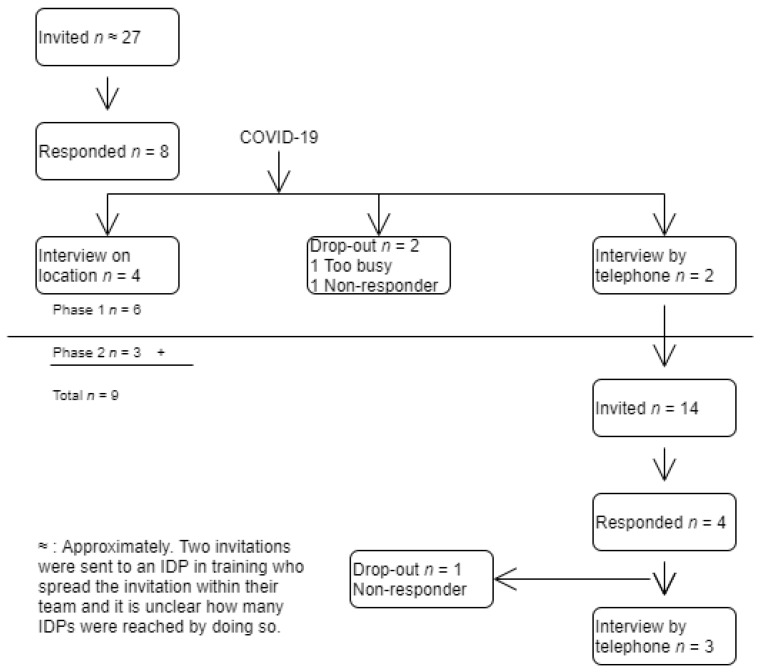
Overview of the recruitment process.

**Figure 2 ijerph-18-10240-f002:**
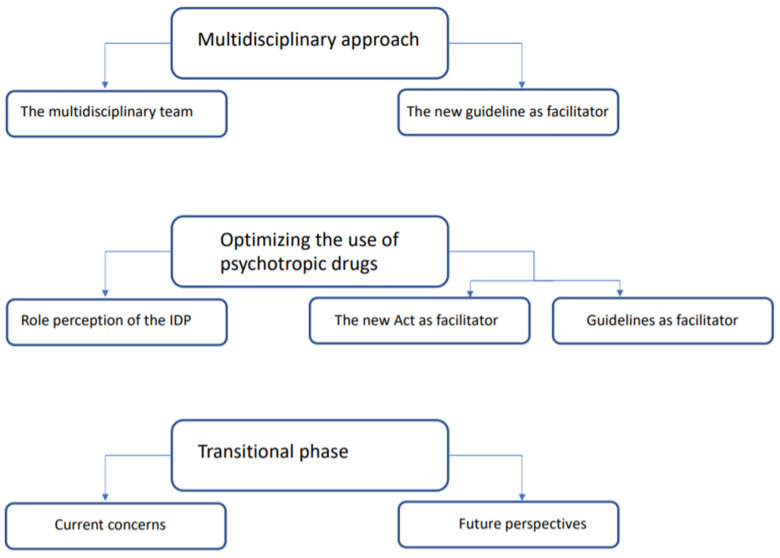
The three major themes and subthemes.

## Data Availability

Data is available on request.

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
