# Peer review of "An Exploratory Study among Intellectual Disability Physicians on the Care and Coercion Act and the Use of Psychotropic Drugs for Challenging Behaviour"

_ijerph, 2021, doi:10.3390/ijerph181910240_

Round 1

Reviewer 1 Report

This was a well designed and interesting study into a legislative reform designed to implement an appropriate response to legitimate concerns noted extensively in connection to the CRPD. Its conclusions are pragmatic, and it should be of relevance to legislators, clinicians, and policymakers in other jurisdictions considering similar law reform processes and their resourcing implications. 

Author Response

Thank you for your kind words in support of our manuscript

Reviewer 2 Report

Dear Authors,

Thank you very much for the opportunity to review this interesting work about a conscious use of psychotropic drugs for challenging behaviors in children with intellectual disability.

In general, I found the topic very interesting, however, a great part of sentences and conclusions should be moderated given the purely qualitative nature of this work. 

Introduction

In general the introduction is linear and clear, however, I would suggest deepening the theoretical part about the challenging behaviors in comorbidity with intellectual disability. Also more explanation about the Care and Coercion Act should be given in this part.

Materials and Methods

Design 

I would suggest adding some more information about the process of thematic content analysis approach.

Participants

The small sample size of this work limits the generalization of the findings and the possibility to make clear statements about the topic. Further, the fact that the physicians are all females limit even more the reliability of the findings reported in this work. In addition to this, also the average work experience was too wide and might constitute another limitation. 

Procedure and Data collection

I would suggest that the authors justify the choice of using open-ended questions during the interviews. I think that these are informative but at the same time the study might have benefitted of more structured questions. 

Data analysis

Line 139: The authors mentioned the agreement for coding the interviews. How was the agreement calculated? Also the numeric value of the agreement should be reported.

Discussion

I would suggest the authors discussing the results also considering that the whole sample was made up by females only. 

In general, discussion should be milder and this study should be considered as a first exploration on these topics.

Author Response

In general, I found the topic very interesting, however, a great part of sentences and conclusions should be moderated given the purely qualitative nature of this work.  

Response: Thank you for your kind words and helpful comments. Below, we respond to all specific points made.  

Introduction. In general the introduction is linear and clear, however, I would suggest deepening the theoretical part about the challenging behaviors in comorbidity with intellectual disability. Also more explanation about the Care and Coercion Act should be given in this part. 

Response: Thank you for your suggestion. We elaborated more on challenging behaviour as well as on the use of drugs. Eg. we added: “It appears that 30 to 50% of children and adolescents with an intellectual disability also have mental disorders (Einfeld, Ellis and Emerson, 2011). Furthermore, there is a greater risk for children and adolescents with an intellectual disability to develop behavioural problems (Dekker, Koot, et al, 2002), such as antisocial and self-absorbed behaviour (Dekker, Nunn et al., 2002).” Furthermore, we added: ''The prevalence of psychotropic drugs prescribed to people with intellectual disabilities in the United Kingdom was found to be respectively 37.7% (Bowring et al., 2017), 49.1% (Henderson et al., 2015) and 49% (Sheehan et al., 2015). Minimizing the use of restrictive practices in the care for people with an intellectual disability has been on the international political agenda for many years (ref Romijn and Frederiks 2012; Sheehan et al., 2017). Policy changes and legislative reforms in various countries aim to reduce both physical and chemical (I.e. by the use of drugs) restraints. Nevertheless, the actual practice is not always in line with these ambitions (Edwards et al., 2020, Deveau & Leitch, 2020).'' 
More information about the Care and Coercion Act is added: '' The new legislation, which replaces the Psychiatric Hospitals (Compulsory Admissions) Act (in Dutch: Wet Bijzondere opnemingen psychiatrische ziekenhuizen, Bopz), is, aiming to regulate the rights of people with an intellectual disability as well as people with a psychogeriatric disorder when receiving involuntary care. 'No compulsion, unless...', is the core of the new Act. Involuntary care can only be applied if there is no other option.''   

Materials and Methods - Design. I would suggest adding some more information about the process of thematic content analysis approach. 

Response: Thank you for your suggestion. We have added more information about our process of the thematic analysis (see Data analysis) and slightly changed the text under Design to be more explicit about the analysis approach. Clarifying the design, we have replaced ''thematic content analysis'' with ''thematic analysis''.  

Participants. The small sample size of this work limits the generalization of the findings and the possibility to make clear statements about the topic. Further, the fact that the physicians are all females limit even more the reliability of the findings reported in this work. In addition to this, also the average work experience was too wide and might constitute another limitation.  

Response: We agree that several facts limit the generalization of our findings. We discuss these limitations in the last paragraph of the discussion. We have rewritten this paragraph slightly and added the limitation of all-female participants.  

Procedure and Data collection. I would suggest that the authors justify the choice of using open-ended questions during the interviews. I think that these are informative but at the same time the study might have benefitted of more structured questions.  

Response: We agree that our choice for using open-ended questions should be motivated. We added our motivation to the text. See Procedure and data collection: “By using open-ended questions, the IDPs could elaborate on points that were important for them, such that all relevant issues were included.”  

Data analysis. Line 139: The authors mentioned the agreement for coding the interviews. How was the agreement calculated? Also the numeric value of the agreement should be reported. 

Response: Given the inductive and explorative character of the study with two researchers no numerical values were calculated, but the coding was followed by thorough discussions throughout the process until consensus was reached. The reliability of the coding was enhanced by respondent validation (Greene & Thorogood, 2018). Please note that to clarify this aspect, changes are made in the method section. 

Discussion. I would suggest the authors discussing the results also considering that the whole sample was made up by females only.  

Response: We added text to the last paragraph of the discussion to discuss the limitations that come from our all-female sample.  

In general, discussion should be milder and this study should be considered as a first exploration on these topics. 

Response: Indeed, the study should be seen as a first exploration, and we have added text to emphasize this. See conclusions: “This study should be considered as a first exploration on these topics.” 

Reviewer 3 Report

Dear Authors, 

Having complete the review of your manuscript, please find below my comments and suggested edits: 

Introduction

  • Lines 70-83. Excellent part of the introduction providing a succinct explanation of the study rationale and the research questions. 

Methods

  • Lines 86-89. You describe your approach to inquiry as exploratory, which is correct. However, you describe your data analysis method as "thematic content analysis" and provide a couple of references that do not refer specifically to this method but to qualitative research methods in general. It would be more appropriate to cite some resources that substantiate your method in line with the use of Braun and Clarke's approach to thematic analysis, which you bring up in lines 135-136. Thematic analysis and content analysis are similar but there are subtle differences that I believe you need to address. In my view, you have used thematic analysis, and not content analysis. This page explains my rationale: https://www.statisticssolutions.com/what-is-the-difference-between-content-analysis-and-thematic-analysis/. On that basis, I would suggest replacing "thematic content analysis" with "thematic analysis" in lines 86 and 87.
  • Lines 111-118. You mention data saturation in this paragraph. For the benefit of readers less used to qualitative research terminology, it would be appropriate to include a sentence describing what this refers to. There are some good ideas here: https://www.ncbi.nlm.nih.gov/pmc/articles/PMC5993836/. 
  • Lines 132-133. Please clarify the following in a short sentence: Inductive reasoning. Please clarify how this type analysis differs from deductive analysis. It starts with the data, from which on works toward broader generalisations and theories. Again, this should be obvious for a qualitative researcher but, sadly, not all readers are duly acquainted with qualitative research terminology. 

Results, Discussion and Conclusion

  • These sections are clear and well written. 

Overall, I found your manuscript interesting and well put together. Your research questions were clear and you answered them appropriately. 

With my suggestions implemented I'll be only too happy to recommend that your paper is accepted for publication. 

All the best,

The Reviewer

Author Response

Introduction 

Lines 70-83. Excellent part of the introduction providing a succinct explanation of the study rationale and the research questions.  

Response: Thank you for your compliments.   

Methods 

Lines 86-89. You describe your approach to inquiry as exploratory, which is correct. However, you describe your data analysis method as "thematic content analysis" and provide a couple of references that do not refer specifically to this method but to qualitative research methods in general. It would be more appropriate to cite some resources that substantiate your method in line with the use of Braun and Clarke's approach to thematic analysis, which you bring up in lines 135-136. Thematic analysis and content analysis are similar but there are subtle differences that I believe you need to address. In my view, you have used thematic analysis, and not content analysis. This page explains my rationale: https://www.statisticssolutions.com/what-is-the-difference-between-content-analysis-and-thematic-analysis/. On that basis, I would suggest replacing "thematic content analysis" with "thematic analysis" in lines 86 and 87. 

Response: Thank you for pointing this out and for providing a helpful explanation on this topic. Indeed, we meant ‘thematic analysis’ and we have taken up your suggestion to remove the word 'content’.   

Lines 111-118. You mention data saturation in this paragraph. For the benefit of readers less used to qualitative research terminology, it would be appropriate to include a sentence describing what this refers to. There are some good ideas here: https://www.ncbi.nlm.nih.gov/pmc/articles/PMC5993836/.  

Response: We agree that an explanation of the term data saturation is a good idea, and we thank you for the link to this interesting article. In the manuscript, we changed the text to: ‘Data saturation was reached after eight interviews, thus, in the eight interview no new information was reported.’ (see Procedure and data collection) 

Lines 132-133. Please clarify the following in a short sentence: Inductive reasoning. Please clarify how this type analysis differs from deductive analysis. It starts with the data, from which on works toward broader generalisations and theories. Again, this should be obvious for a qualitative researcher but, sadly, not all readers are duly acquainted with qualitative research terminology.  

Response: We have added the following sentence at this point in the text: ‘The thematic analysis was data driven and explorative, rather than theoretically driven (deductive analysis), and themes were identified inductively, hence, working toward broader generalisations and theories.’ 

Results, Discussion and Conclusion 

These sections are clear and well written.  

 Overall, I found your manuscript interesting and well put together. Your research questions were clear and you answered them appropriately.  

With my suggestions implemented I'll be only too happy to recommend that your paper is accepted for publication.  

Response: We thank you for your compliments and helpful suggestions with which we were able to improve our manuscript.  

Reviewer 4 Report

Some comments are suggested:

  • The abstract must be structured with the sections of the manuscript specifying introduction, clearly summarized methodology, results, discussion and main conclusion.
  • It would be interesting if the keywords are DeCS / MeSH descriptors.
  • In the introduction, the phenomenon is adequately described, but it would be necessary to include some description of similar studies, of the perception and experience in other places, with similar plans or phenomena, of how the pharmacological approach of these patients is.
  • The research questions should not be included at the end of the introduction but in the methodology section
  • Would you specify what type of qualitative approach did you use: phenomenological as it appears to be or other as entnographic?
  • What have been the segmentation variables? the years of experience? Why? The rest of the characteristics, such as the environments where they worked, were also considered as an aspect that modified the discourse? Try to clarify these aspects in the methodology
  • The methodology should include a subsection of strategies of methodological rigor such as triangulation, credibility, transferability, etc.
  • The discussion is interesting, but I think it is necessary to connect with similar qualitative studies to ensure the transferability of the results.
  • The last paragraph of the discussion seems to be the limitations of the study. It should be clearly stated.

Author Response

The abstract must be structured with the sections of the manuscript specifying introduction, clearly summarized methodology, results, discussion and main conclusion. 

Response: We have followed the author instructions (https://www.mdpi.com/journal/ijerph/instructions) where it was stated that headings should not be used in the abstract. We have tried to follow the style of a structured abstract, but without headings.  

It would be interesting if the keywords are DeCS / MeSH descriptors. 

Response: Thank you for your suggestion. We revised the keywords.  

In the introduction, the phenomenon is adequately described, but it would be necessary to include some description of similar studies, of the perception and experience in other places, with similar plans or phenomena, of how the pharmacological approach of these patients is. 

Response: We added several sentences to describe the broader international perspective regarding the use of restrictive practices, in particular the use of psychotropic drugs in case of challenging behaviour. We added among others: ‘The prevalence of psychotropic drugs prescribed to people with intellectual disabilities in the United Kingdom was found to be respectively 37.7% (Bowring et al., 2017), 49.1% (Henderson et al., 2015) and 49% (Sheehan et al., 2015).’ (See Introduction).  

The research questions should not be included at the end of the introduction but in the methodology section 

Response: The location of the research questions may perhaps vary per research field or journal. In the fields of social sciences and medicine it is common to specify the research questions at the end of the introduction. Since one of the other reviewers indicated this paragraph as one of the particular good parts of the introduction, we hope that you can agree with leaving the research questions in the introduction.  

Would you specify what type of qualitative approach did you use: phenomenological as it appears to be or other as entnographic? 

Response: Thank you for your remark. We added that it is a phenomenological study (see Design) 

What have been the segmentation variables? the years of experience? Why? The rest of the characteristics, such as the environments where they worked, were also considered as an aspect that modified the discourse? Try to clarify these aspects in the methodology 

Response: In our sample we have tried to choose the group of IDPs as diverse as possible, also with respect to years of experience, such as to end up with a sample that is representative for the population of (female) IDPs in the Netherlands.  

To be more clear about the variables we revised the text in the method section to: “We strived for a broad range with respect to age and years of work experience and aimed for variety with respect to work environment, in order to end up with a sample that is representative for our population of IDPs in the Netherlands.” (see Procedure and data collection).  

The methodology should include a subsection of strategies of methodological rigor such as triangulation, credibility, transferability, etc. 

Response: For a rigor study special attention was given to the credibility and transferability of the study. We made this more explicit by mentioning these terms and adding text about this in the section 'Procedure and data collection'. We added: ‘The diversity of the sample contributes to the transferability of the study. ' and ‘'The fact that data have been collected until data saturation contributes to the credibility of the study.'  

However, more could have been done concerning triangulation, for example by adding a questionnaire to the study. We will consider this in future research.  

The discussion is interesting, but I think it is necessary to connect with similar qualitative studies to ensure the transferability of the results. 

Response: We agree that, ideally, we compare our study to similar qualitative studies in the discussion. We have tried to connect to relevant studies and news items. However, since our study is concerned with a specific event (a new Law) and the consequences for a relatively small group of people (the IDPs), no similar studies can be found. We therefore hope that our study triggers new research, for example to see whether the opinions of the IDPs are shared by other professionals, as we write in the conclusions of our manuscript.  

The last paragraph of the discussion seems to be the limitations of the study. It should be clearly stated. 

Response: We agree that it would be better to make the limitations of the study more explicit. We revised this paragraph to make this more clear.  

Round 2

Reviewer 2 Report

The authors responded appropriately to all questions.